# Online Hybrid Neural Network for Stock Price Prediction: A Case Study of High-Frequency Stock Trading in the Chinese Market

**Chengyu Li** , **Luyi Shen** **and Guoqi Qian** *

School of Mathematics and Statistics, The University of Melbourne, Parkville, VIC 3010, Australia
* Correspondence: qguoqi@unimelb.edu.au

**Abstract:** Time-series data, which exhibit a low signal-to-noise ratio, non-stationarity, and non-linearity, are commonly seen in high-frequency stock trading, where the objective is to increase the likelihood of profit by taking advantage of tiny discrepancies in prices and trading on them quickly and in huge quantities. For this purpose, it is essential to apply a trading method that is capable of fast and accurate prediction from such time-series data. In this paper, we developed an online time series forecasting method for high-frequency trading (HFT) by integrating three neural network deep learning models, i.e., long short-term memory (LSTM), gated recurrent unit (GRU), and transformer; and we abbreviate the new method to online LGT or O-LGT. The key innovation underlying our method is its efficient storage management, which enables super-fast computing. Specifically, when computing the forecast for the immediate future, we only use the output calculated from the previous trading data (rather than the previous trading data themselves) together with the current trading data. Thus, the computing only involves updating the current data into the process. We evaluated the performance of O-LGT by analyzing high-frequency limit order book (LOB) data from the Chinese market. It shows that, in most cases, our model achieves a similar speed with a much higher accuracy than the conventional fast supervised learning models for HFT. However, with a slight sacrifice in accuracy, O-LGT is approximately 12 to 64 times faster than the existing high-accuracy neural network models for LOB data from the Chinese market.

**Keywords:** high-frequency; limit order book; online fast prediction; hybrid neural network



## 1. Introduction

More and more investment institutions have entered the trading practice as the financial markets have significantly grown in recent years. This has led to a rapid increase in the amount of financial time-series data generated through high-frequency trading on the financial markets, presenting both opportunities and challenges for researchers to tackle. Our study in this paper focuses on analyzing the time-series data of limit order books (LOBs) for high-frequency trading (HFT). LOBs are records of outstanding limit orders maintained by the security specialists who work at the exchange. A limit order is a type of order to buy or sell a security at a specific price or higher. LOBs can be regarded as financial time series that reflect expected price levels for traders. By using a limit order book, traders can specify the exact price at which they want to buy or sell a security, such as an asset stock, so that the involved risk can be properly managed and the prospective returns can be maximized. However, LOBs are characterized by a low signal-to-noise ratio, non-stationarity, and non-linearity (Henrique et al. 2019), making it a challenge to effectively and efficiently analyze them.

Stock (or more generally, security) price prediction is a key task in analyzing LOB data, which helps investors develop trading strategies and select investment portfolios that are more likely to produce high returns with low risks. Accurate forecasting of stock prices requires a robust and efficient model. Despite the abundance of research in this field,

the challenges associated with the speed of computing such models remain. The main objective of our paper was to develop a fast online hybrid neural network model to predict stock prices.

To prepare for this development, a brief review of the current methods for stock price prediction is presented in the following. Overall, most of these methods fall into three categories: statistical parametric models, machine learning techniques, and deep learning approaches.

Regarding the statistical parametric models for stock price prediction, Cenesizoglu et al. (2016) extracted the informative variables that characterize LOBs, so as to establish a vector auto-regressive (VAR) model to analyze how various features of the LOBs affect prices. Mondal et al. (2014) evaluated the accuracy and variability of stock price forecasts using an auto-regressive integrated moving average (ARIMA) model. They employed the model selection criterion AICc to estimate the optimum ARIMA model and also analyzed the impact of altering the time frame of historical data on prediction accuracy. Tran et al. (2017) employed multi-linear discriminant analysis (MDA) to forecast large-scale mid-price movements through high-frequency limit order book data. Catania et al. (2022) introduced a multi-variate model for discrete high-frequency stock price changes using a hierarchical hidden Markov model based on the Skellam distribution, which accounts for the large proportion of zero returns and the co-staleness phenomenon. Although statistical parametric models are computationally efficient, they have limitations when applied to complex stock prices data and may not yield the desired results due to their strong dependence on assumptions that may not be met by these data.

A data-driven machine learning model is not typically constrained by assumptions. Rather, it uses the data themselves to identify patterns and relationships that can inform predictions. Yun et al. (2021) proposed a system for predicting the direction of stock price movements that emphasizes an enhanced feature engineering process. The system utilizes a hybrid of genetic algorithms and extreme gradient boosting (GA-XGBoost) to optimize the selection of features used in the prediction. Kercheval and Zhang (2015) used a multi-class support vector machine to capture the dynamics of high-frequency LOBs, which automatically predicts mid-price movement and other indicators in real time. Previous works in machine learning for stock price prediction have highlighted the importance of extracting relevant features from the underpinning big data for prediction.

Deep learning is a branch of machine learning that uses neural networks, each with multiple layers, to analyze data. These layers sequentially transform the raw data into informative statistics, allowing the model to extract important features and patterns from the raw data. A convolutional neural network (CNN) is a typical example. Tsantekidis et al. (2017) applied a deep learning approach that forecasts stock price movements by utilizing a CNN. Experiments show that the results of the CNN outperform many other machine learning models such as support vector machines. However, compared to other network structures, a CNN is relatively unsophisticated and underperforms in analyzing high-frequency trading data.

Recurrent neural networks (RNNs), such as long short-term memory (LSTM) (Hochreiter and Schmidhuber 1997) and gated recurrent unit (GRU) (Cho et al. 2014), have been widely used to predict stock prices. These architectures are well suited for time-series data, such as stock data, because they have the ability to keep previous inputs in their memory, which is important for incorporating the temporal dependencies from the inputs into the prediction process. Chen and Zhou (2020) proposed a stock prediction model that combines genetic algorithm feature selection with LSTM neural networks, demonstrating improved performance over benchmark models. As the complexity of financial data increases, more advanced neural network architectures have emerged to address these challenges. The transformer (Vaswani et al. 2017) architecture, which utilizes a self-attention mechanism to weigh the importance of various input sequences, has been particularly useful for stock prediction, as it enables the model to weigh the importance of all financial indicators from the past, such as previous stock prices and volumes. Ding et al. (2020) developed a novel

transformer-based approach for stock movement prediction, introducing enhancements such as multi-scale Gaussian prior, orthogonal regularization, and trading gap splitter. These improvements increased locality, reduced redundant head learning in multi-head self-attention, and captured hierarchical features in high-frequency financial data.

Hybrid neural networks, which combine different types of neural networks, can have better overall performance in stock prediction because they take advantage of the strengths of their respective architectures. Zhang et al. (2019) proposed deep convolutional neural networks for limit order books (DeepLOB). Three building blocks make up the network architecture of DeepLOB: convolutional layers, parallel inception layers, and an LSTM layer. Zhang and Zohren (2021) proposed DeepAcc for LOBs, which combines DeepLOB with a hardware acceleration mechanism for performing stock prediction. Both DeepLOB and DeepAcc utilize a CNN model as the encoder, which transforms the stock data into a vector. This is followed by a decoder that produces the final output.

While the accuracy of price prediction is undoubtedly a key performance indicator, the issue of computing speed by these deep learning methods is largely ignored in assessing the methods' performance in the literature. In fact, in high-frequency trading, speed is crucial to a number of strategies, e.g., cross-market arbitrage and market making. Baron et al. (2019) discovered that variations in the relative latency can have a significant impact on the trading performance of HFT firms when they investigated the competition among these firms. Therefore, it is necessary to improve the prediction speed of their methods for investors to obtain more profits without undue risks.

Motivated by the above review and discussion, we propose an online hybrid neural network method for predicting stock prices based on high-frequency LOB time-series data, where we focus on achieving optimal computing speed while maintaining a high prediction accuracy and feasible computing memory. Our proposed method was developed by integrating the three neural network deep learning models LSTM, GRU, and transformer into an online architecture; hence, it is named online LGT or O-LGT. The key innovation underlying O-LGT is its efficient storage management, enabling super-fast computing. Specifically, when computing the stock forecast for the immediate future, we only use the output calculated from the previous trading data (rather than the previous trading data themselves) together with the current trading data. Thus, the computing only involves updating the current data in the process. Details of the method are presented in Section 3.

Comparisons of our proposed method with the currently available stock price prediction methods reviewed above are also presented in this paper. For the Chinese stock market LOB data that are used in this paper, we found the best of the reviewed methods typically took at least 2.21 ms of computing time to reach the level of accuracy achieved by the O-LGT method. On the other hand, on the same computer with the same computing power, it typically took O-LGT 0.0579 ms of computing time to reach the same level of accuracy, approximately 40 times faster than the best reviewed method. More details of the comparison are presented in Section 4. The improvement of the computing speed has significant implications for the traders in HFT, because it gives them more time to make decisions and execute orders than their competitors.

This paper is structured as follows. In Section 2, we describe the LOB data used in our work in this paper. Next, we present the development of the methods in Section 3, including the problem statement, the methods of RNN, LSTM, GRU, and transformer, and the framework of our O-LGT method. Section 4 presents the details and results of our experiments. Finally, we conclude the paper with a summary in Section 5.

## 2. Data and Materials

### 2.1. High-Frequency Limit Order Book Data

As introduced in Gould et al. (2013), more than 50% of the global financial markets use limit order books (LOBs) to match sellers and buyers. First, we present some definitions of a limit order book (LOB). A limit order is a contract to buy or sell a limited quantity of shares at a specific price (Gould et al. 2013). Specifically, a sell limit order ensures that

the seller sells a specified amount of shares of a stock at a price no less than the specified ask price. Contrarily, a buy limit order ensures that the buyer buys a specified amount of shares of a stock for no more than the specified bid price. Accordingly, orders in an LOB can either be ask orders or bid orders.

At a given time $t$, let $\mathbf{p}_{ask}(t)$ and $\mathbf{v}_{ask}(t)$ represent the column vectors of prices and volumes of all the ask orders, and let $\mathbf{p}_{bid}(t)$ and $\mathbf{v}_{bid}(t)$ represent the column vectors of prices and volumes of all the bid orders. In addition, let $p_{ask}^{(1)}(t)$ be the lowest available sell price in the ask orders and $p_{bid}^{(1)}(t)$ be the highest available buy price in the bid orders. When $p_{ask}^{(1)}(t) < p_{bid}^{(1)}(t)$, the orders are executed and the traded assets are exchanged between the investors on a first-come, first served basis.

Figure 1 gives an illustration of LOBs in trading at a given time stamp, where the five best-priced buy LOB bars and the five best-priced sell LOB bars are sorted according to the price, and the height of each bar represents the volume of the associated LOB. Here, the highest bid price $p_{bid}^{(1)}(t)$ is greater than the lowest ask price $p_{ask}^{(1)}(t)$, thus a transaction of size equal to the volume of the lowest sell limit order is immediately completed between the lowest sell limit order and the highest buy limit order. Thereafter, the highest buy limit order still has some volume left to be executed because its volume is greater than that of the lowest sell limit order. In cases when multiple sell and buy limit orders partially match, the transactions will be carried out on the first-come, first served basis until no more matches exist in the market.

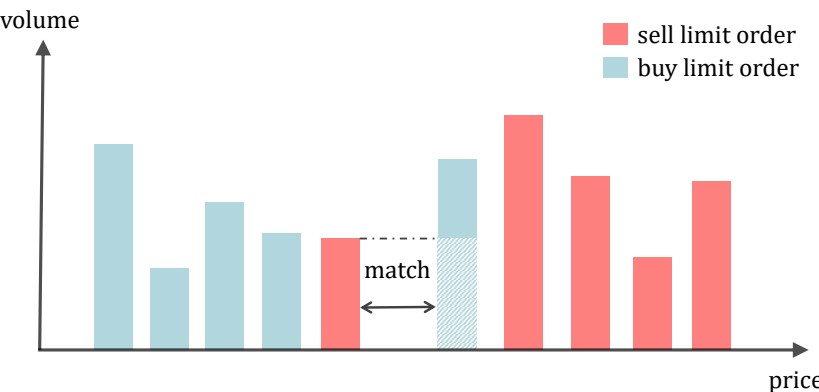

**Figure 1.** An illustration of LOBs in trading at a given time stamp, where the bid and ask orders are sorted by price. When a bid order is priced higher than an ask order, the two are automatically matched and put into execution.

In HFT markets, the LOBs are traded electronically at high frequency, in large numbers, and at a huge volume. To facilitate investors to receive large profits with low risks from HFT, it is important to continuously provide them with timely information of accurate stock price predictions. Thus, a key task in HFT of LOBs is developing an online stock price prediction model based on the LOB data.

*2.2. LOBs in the Chinese Market*

The LOB data to be analyzed in this paper came from the CSI Smallcap 500 Index in the Chinese market. The observed data include trading records of 100 stocks. These records were updated every 3 s for a total of 22 trading days in November 2021. Each trading day comprises 3 h and 57 min of active trading time, resulting in a total of 4740 records per day.

This paper uses 27 features to summarize each trading record, as described in Table 1. Among them, the first 26 features serve as inputs, while the last is the percentage change in stock price over a 5-min period, which serves as the output. Specifically, these input features include the highest five prices bid $p_{bid}^{(1)}(t), \cdots, p_{bid}^{(5)}(t)$ and the corresponding LOB volumes $v_{bid}^{(1)}(t), \cdots, v_{bid}^{(5)}(t)$; the lowest five ask prices $p_{ask}^{(1)}(t), \cdots, p_{ask}^{(5)}(t)$ and the corresponding

LOB volumes $v_{ask}^{(1)}(t), \cdots, v_{ask}^{(5)}(t)$; the average transaction price over the last three seconds $p_{tra}^{avg}(t)$; the total ask volumes $v_{ask}^{(all)}(t)$; the total bid volumes $v_{bid}^{(all)}(t)$; the average ask price $p_{ask}^{avg}(t)$; the average bid price $p_{bid}^{avg}(t)$; the latest transaction price $p_{last}(t)$.

**Table 1.** Description of the features.

| Features | Description |
|---|---|
| $p_{bid}^{(1)}(t), \cdots, p_{bid}^{(5)}(t)$ | Highest five prices bid at time $t$ |
| $v_{bid}^{(1)}(t), \cdots, v_{bid}^{(5)}(t)$ | Corresponding LOB bid volumes of the highest five prices bid at time $t$ |
| $p_{ask}^{(1)}(t), \cdots, p_{ask}^{(5)}(t)$ | Lowest five ask prices at time $t$ |
| $v_{ask}^{(1)}(t), \cdots, v_{ask}^{(5)}(t)$ | Corresponding LOB ask volumes of the lowest five ask prices at time $t$ |
| $v_{ask}(t)$ | Total ask volumes at time $t$ |
| $v_{bid}(t)$ | Total bid volumes at time $t$ |
| $p_{avg}(t)$ | Average transaction price over the last 3 s |
| $p_{ask}(t)$ | Average ask price over the last 3 s |
| $p_{bid}(t)$ | Average bid price over the last 3 s |
| $p_{last}(t)$ | Latest transaction price at time $t$ |
| $p(t)$ | A stock price at time $t$ |

## 3. Methods

### 3.1. Problem Statement

The problem tackled in this paper relates to developing an online hybrid neural network model for continuous prediction of stock prices based on high-frequency LOB data. Let $p(T)$ be the price of a stock at time $T$, with $T = 1, 2, \cdots$ denoting the number of time units passed from the beginning of trading on each day. For the Chinese LOB data, the time unit is 3 s. Predicting $p(T)$ is equivalent to predicting the target variable $y_T$, defined as $y_T = \frac{p(T) - p(T-h)}{p(T-h)}$, which represents the percentage change in the stock price between time $T$ and $h$ units of time earlier. In the Chinese HFT market, one typically predicts $y_{T+h}$ at time $T$ with $h = 100$, i.e., predicts the price 5 min forward. If $\mathbf{x}_t$ is denoted as the $J = 26$ features of the Chinese LOB data recorded at time $t$ as defined in Section 2.2, then the prediction $\hat{y}_T$ of $y_T$ by the LOB features recorded in the previous $s$ time steps can be generically formulated as the following:

$$\hat{y}_T = \hat{F}(\mathbf{x}_{T-1}, \mathbf{x}_{T-2}, \cdots, \mathbf{x}_{T-s}) \tag{1}$$

where $F(\cdot)$ denotes a generic neural network and $\hat{F}(\cdot)$ is an estimate of $F(\cdot)$ obtained from the training data. The best predictions $\hat{y}_T$ across all values of $T$ are to be computed by minimizing $D(y, \hat{y})$, where $D(\cdot, \cdot)$ is a customized discrepancy function that measures the proximity of an estimate to its actual value.

We developed an online hybrid neural network method to formulate and optimally estimate $F(\cdot)$. This method is named O-LGT since it is in the form of a general recurrent neural network (RNN) containing multiple latent layers, which are specified by the long short-term memory (LSTM) model, the gated recurrent unit (GRU) model, and the transformer model in that sequence. In addition, the method is implemented in an online way, i.e., it is updated once the time moves forward by one unit. In order to give a detailed description of O-LGT, we first review all of its layers in the following.

### 3.2. Recurrent Neural Network (RNN)

First, recall that recurrent neural networks (RNNs) are a type of neural network that are designed to process sequential data, such as time-series data. An RNN contains a hidden compartment that inputs information on the data from both the previous and the current time steps, and outputs predictions for future time steps. This hidden compartment is updated once at each new time step, by passing the input through to generate the output.

The structure of an RNN is cyclical, meaning that the same computation is performed at each step using the same parameters, which is why it is called recurrent. The architecture of RNNs can be unrolled to show the sequence of computations that occur over time.

Figure 2 is a typical RNN structure diagram, where $\mathbf{h}_t$, the output vector of the hidden layer at time $t$, depends not only on the input vector $\mathbf{x}_t$ at time $t$, but also on $\mathbf{h}_{t-1}$. The $\mathbf{h}_t$ is calculated using an activation function $f(\cdot)$ as follows:

$$\mathbf{h}_t = f(U\mathbf{x}_t + W\mathbf{h}_{t-1}) \tag{2}$$

where $U$ is the input layer weight matrix and $W$ represents the hidden layer weight matrix. The RNN is initialized at time $t = 1$ with

$$\mathbf{h}_1 = f(U\mathbf{x}_1 + W\mathbf{h}_0) \quad \text{and} \quad \mathbf{y}_1 = g(V\mathbf{h}_1) \tag{3}$$

where $g(\cdot)$ is the activation function for the output layer and $V$ is the associated weight matrix. It then proceeds with the following operations at each time stamp $t = 2, 3, \cdots$:

$$\mathbf{h}_t = f(U\mathbf{x}_t + W\mathbf{h}_{t-1}) \quad \text{and} \quad \mathbf{y}_t = g(V, \mathbf{h}_t) = g(V\mathbf{h}_t) \tag{4}$$

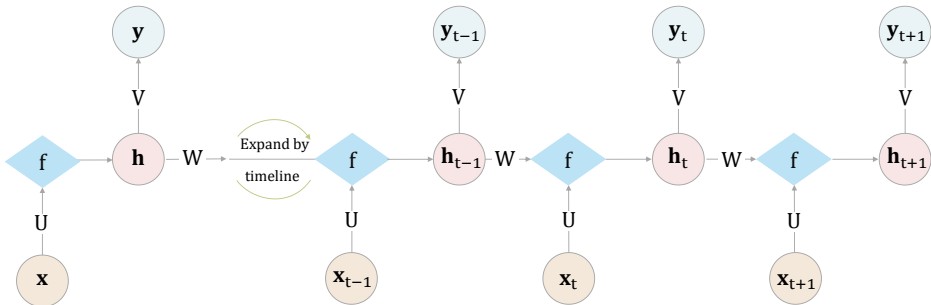

**Figure 2.** The structure of a recurrent neural network. $\mathbf{x}_t$ is the input at time $t$; $\mathbf{h}_t$ denotes the output of hidden layer at time $t$; $\mathbf{y}_t$ is final output at time $t$; $U$ represents the input layer weight matrix; $W$ represents the hidden layer weight matrix; $V$ represents the output layer weight matrix; $f(\cdot)$ is the activation function.

### 3.3. Long Short-Term Memory (LSTM) and Gated Recurrent Unit (GRU)

Hochreiter and Schmidhuber (1997) introduced the long short-term memory (LSTM) architecture to address the problem of vanishing gradients in traditional recurrent neural networks (RNNs) when trying to model long-term dependencies in sequential data.

The core of LSTM is a cell consisting of an input gate $\mathbf{i}_t$, a forget gate $\mathbf{f}_t$, a candidate state $\widetilde{\mathbf{c}}_t$, and an output gate $\mathbf{o}_t$, normally in the form of column vectors, as shown in the left panel of Figure 3. The forget gate $\mathbf{f}_t$ determines the information to be directly passed to the cell state output $\mathbf{c}_t$, the input gate $\mathbf{i}_t$ determines the information to be further used in the cell together with the candidate state $\widetilde{\mathbf{c}}_t$, and the output gate $\mathbf{o}_t$ together with the cell state output $\mathbf{c}_t$ gives the hidden layer output $\mathbf{u}_t$.

The "long" in LSTM stands for the capability of the model to retain information for a long period of time. The memory cells in LSTMs, which can keep information for multiple time steps, enable the model to effectively identify long-term dependencies in sequential data. The "short" in LSTM stands for the model's ability to discard irrelevant information promptly. An LSTM model uses the forget gate and input gate to appropriately store and retrieve selected information from the memory cells.

At each time $t$, an LSTM model with a hidden layer dimension $q_L$ loads the $J \times 1$ feature vector $\mathbf{x}_t$, the $q_L \times 1$ hidden layer output vector $\mathbf{u}_{t-1}$ at the previous time stamp, and the $q_L \times 1$ cell state vector $\mathbf{c}_{t-1}$ at the previous time stamp as the input. It then uses certain activation functions to generate $[\mathbf{f}_t, \mathbf{i}_t, \widetilde{\mathbf{c}}_t, \mathbf{o}_t]$, which is a $q_L \times 4$ matrix. Finally, it outputs the updates $\mathbf{u}_t$ and $\mathbf{c}_t$. An update in LSTM at time $t$ can be expressed as follows:

$$\mathbf{i}_t = \sigma\left(W_i\begin{bmatrix}\mathbf{x}_t \\ \mathbf{u}_{t-1}\end{bmatrix} + \mathbf{b}_i\right)$$

$$\widetilde{\mathbf{c}}_t = \tanh\left(W_c\begin{bmatrix}\mathbf{x}_t \\ \mathbf{u}_{t-1}\end{bmatrix} + \mathbf{b}_c\right)$$

$$\mathbf{f}_t = \sigma\left(W_f\begin{bmatrix}\mathbf{x}_t \\ \mathbf{u}_{t-1}\end{bmatrix} + \mathbf{b}_f\right) \tag{5}$$

$$\mathbf{o}_t = \sigma\left(W_o\begin{bmatrix}\mathbf{x}_t \\ \mathbf{u}_{t-1}\end{bmatrix} + \mathbf{b}_o\right)$$

$$\mathbf{c}_t = \mathbf{i}_t \odot \widetilde{\mathbf{c}}_t + \mathbf{f}_t \odot \mathbf{c}_{t-1}$$

$$\mathbf{u}_t = \mathbf{o}_t \odot \tanh(\mathbf{c}_t)$$

where $W_i$, $W_f$, $W_o$, and $W_c$ are the $q_L \times (J + q_L)$ weight matrices of the input gate, the forget gate, the output gate, and the cell state, respectively; $\mathbf{b}_i$, $\mathbf{b}_f$, $\mathbf{b}_o$, and $\mathbf{b}_c$ are the $q_L \times 1$ bias vector parameters of the input gate, the forget gate, the output gate, and the cell state, respectively. In addition, $\odot$ is the Hadamard product operation. Moreover, $\sigma(\cdot)$ and $\tanh(\cdot)$ are activation functions defined as

$$\sigma(x) = \frac{1}{1 + \exp(-x)} \quad \text{and} \quad \tanh(x) = \frac{\exp(x) - \exp(-x)}{\exp(x) + \exp(-x)} \tag{6}$$

The GRU model is a simplified version of the LSTM model and was introduced by Cho et al. (2014). Compared with LSTM, the GRU has fewer parameters and is computationally less expensive to train, while still being capable of capturing long-term dependencies in sequential data and robust against overfitting. The main difference between LSTM and the GRU is that the GRU combines the omission and input gates of LSTMs into a single update gate $\mathbf{z}_t$. The GRU structure is depicted in the right panel of Figure 3. The core of the GRU is a cell consisting of a reset gate $\mathbf{r}_t$, an update gate $\mathbf{z}_t$, and an output $\widetilde{\mathbf{v}}_t$.

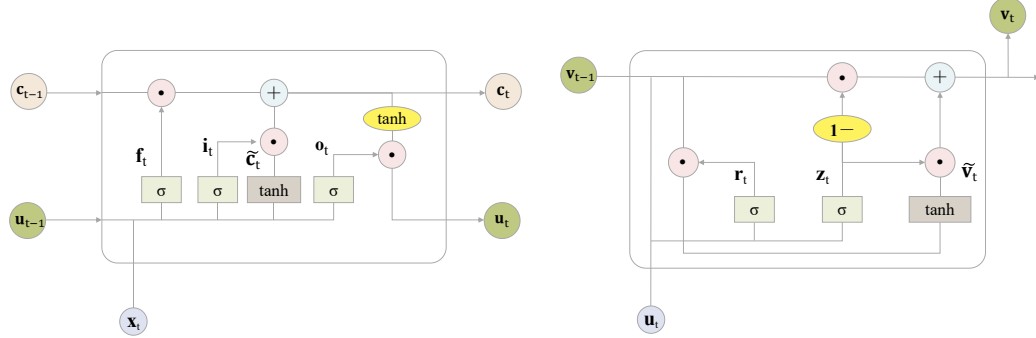

**Figure 3.** The **left** panel shows the structure of long short-term memory, and the **right** panel shows the structure of the gated recurrent unit.

The "Gated" in GRU refers to the use of gates to control the flow of information in and out of the hidden layer. "Recurrent" refers to the fact that the GRU is a type of recurrent neural network (RNN) that processes sequential data by passing information from one time step to the next.

At each time $t$, a GRU model with a hidden layer dimension $q_G$ loads the $q_L \times 1$ LTSM hidden layer output $\mathbf{u}_t$ and the $q_G \times 1$ GRU output vector $\mathbf{v}_{t-1}$ at the previous time stamp as the input. It then uses the activation functions $\sigma(\cdot)$ and $\tanh(\cdot)$ to compute $[\mathbf{r}_t, \mathbf{z}_t, \widetilde{\mathbf{v}}_t]$, which is a $q_G \times 3$ matrix. Finally, it generates the updated GRU output vector $\mathbf{v}_t$. An update in the GRU at time $t$ can be expressed as follows:

$$\mathbf{v}_t = (1 - \mathbf{z}_t) \odot \tilde{\mathbf{v}}_t + \mathbf{z}_t \odot \mathbf{v}_{t-1}$$

$$\tilde{\mathbf{v}}_t = \tanh\left(W_v \begin{bmatrix} \mathbf{u}_t \\ \mathbf{r}_t \odot \mathbf{v}_{t-1} \end{bmatrix} + \mathbf{b}_v\right)$$

$$\mathbf{r}_t = \sigma\left(W_r \begin{bmatrix} \mathbf{u}_t \\ \mathbf{v}_{t-1} \end{bmatrix} + \mathbf{b}_r\right) \tag{7}$$

$$\mathbf{z}_t = \sigma\left(W_z \begin{bmatrix} \mathbf{u}_t \\ \mathbf{v}_{t-1} \end{bmatrix} + \mathbf{b}_z\right)$$

where $W_v$, $W_z$, and $W_r$ are the $q_G \times (q_L + q_G)$ weight matrices of the GRU output, the update gate, and the reset gate, respectively; and $\mathbf{b}_v$, $\mathbf{b}_r$, and $\mathbf{b}_z$ are the $q_G \times 1$ bias vector parameters of the GRU output, the reset gate, and the update gate, respectively.

Note that the reset gate $\mathbf{r}_t$ is used to control the extent to which the output information of the previous time moment is ignored. Typically, the smaller the value of the reset gate, the more likely it is ignored. In addition, a larger value of the update gate $\mathbf{z}_t$ indicates that the neural unit at the current time moment is less influenced by the output information of the neural unit from the previous time moment.

*3.4. Transformer*

After being processed by LSTM and GRU, the multi-headed transformer aids in extracting useful information regarding the interactions in outputs between various time steps.

There are significant differences between the transformer (Vaswani et al. 2017) and the traditional RNN model, in that the attention mechanism in the former completely determines the structure of the entire network. Attention allows the model to focus on specific parts of the input by assigning different weights to different positions in the input sequence. This is in contrast to the traditional RNNs, which use the same weights for all positions in the input sequence.

The transformer model uses the encoder–decoder architecture that is most commonly used in Neuro-linguistic programming. This architecture provides an effective way to handle long sequence data (Bahdanau et al. 2014). In the transformer model, the encoder has four layers: The first layer uses a multi-head attention mechanism (multi-head attention) to assign multiple sets of different attention weights to the model for extending the model's ability to focus on different locations, thus capturing richer information than otherwise. The second layer is the summation and normalization layer, where the summation, also named the residual connection, adds the interim output of the layer to its input before being normalized to produce the layer's final output. The second layer passes the information from the previous layer to the next layer without differences to solve the gradient disappearance problem more quickly. The third layer is a feed forward neural network (FNN) layer. The fourth layer then goes through another summation and normalization layer to generate the intermediate semantic coding vector and transmit it to the decoder. The decoder has six layers, similar to the encoder structure, but the first layer is a multi-headed attention layer with a MASK (masking) operation, because at output time $t$, the information at time $t + 1$ is not available, so the output of the decoder needs to be shifted right and the subsequent items are masked for prediction. Finally, the decoder goes through linear regression and the Softmax layer to output the final prediction result. Figure 4 displays the block diagram for the transformer model.

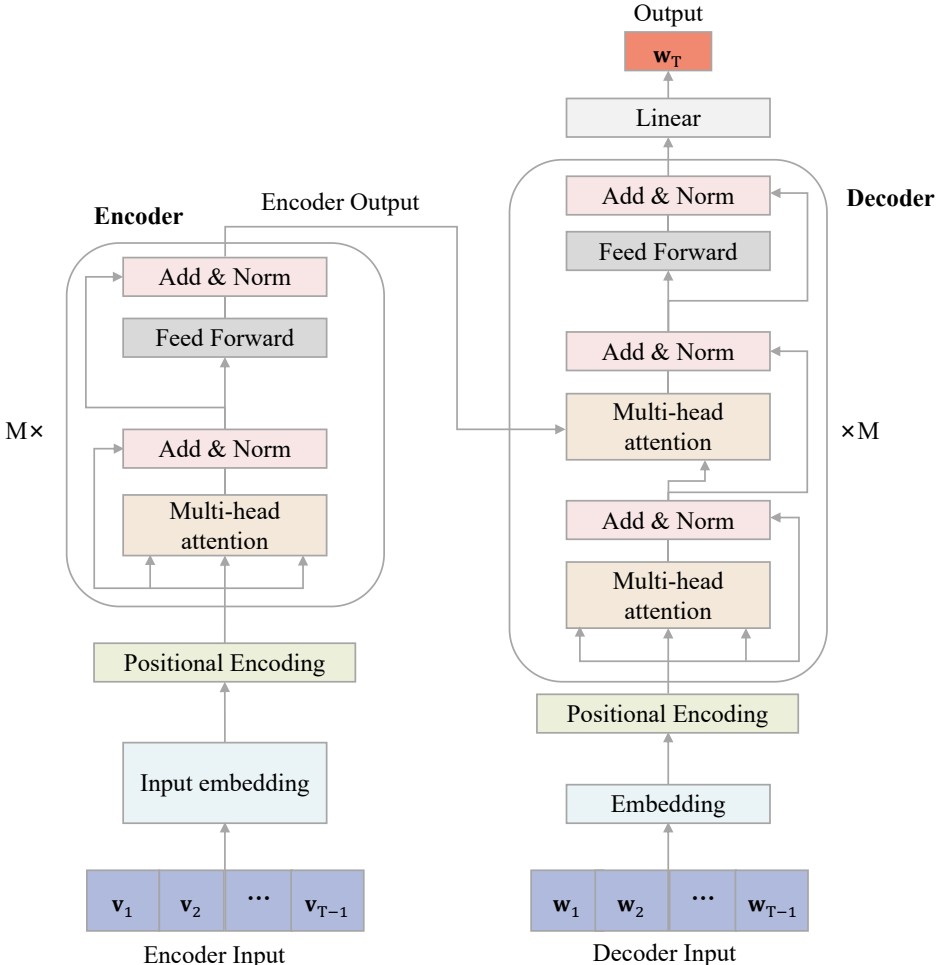

**Figure 4.** Illustration of a transformer. The dimension of $\mathbf{v}_i$ is the input dimension of the transformer and the dimension of $\mathbf{w}_i$ is the hidden layer dimension of the transformer.

### 3.5. Online LGT (O-LGT)

In this section, we present the details of our developed stock price prediction method, O-LGT, having reviewed the necessary RNNs. We will first provide a brief description of the sweep matrix operator, which serves as the core computing engine for O-LGT. We will then describe the framework of O-LGT and its practical implementation. Finally, we will discuss the standardization and transformation of input data before using them in O-LGT execution.

#### 3.5.1. Sweep Operator

Computations involved in all neural network layers in O-LGT are essential for solving weighted regression normal equations $X^T W X \hat{\beta} = X^T W y$. The sweep matrix operator (Beaton 1964) provides a very efficient method to solve these normal equations. Following Goodnight (1979), a square matrix $M = (m_{ij})$ is said to have been swept on the $k$th row and column (or $k$th pivotal element) when it has been transformed into a matrix $N = (n_{ij})$, such that

$$
\begin{aligned}
n_{kk} &= 1/m_{kk} \\
n_{ik} &= -m_{ik}/m_{kk} \quad i \neq k \\
n_{kj} &= m_{kj}/m_{kk} \quad j \neq k \\
n_{ij} &= m_{ij} - m_{ik}m_{kj}/m_{kk} \quad i, j \neq k.
\end{aligned}
\tag{8}
$$

Via a sweep operation on each pivotal element of $M$, each element of $M$ is updated, essentially by one division operation. By applying sweep operations on all pivotal elements of $X^TWX$, the normal equation $X^TWX\hat{\beta} = X^TWy$ can be solved in approximately $O(2||X^TWX||_0^2)$ divisions, with $||X^TWX||_0$ being the number of elements in matrix $X^TWX$, which is the same computing complexity as that involved in the classical Gauss–Jordan elimination operator. However, when an extra column of data is augmented to $X$, the resultant new normal equation can be solved by updating the process of solving the previous normal equation by applying just one more sweep operation on the new pivotal element, which only takes $O(||X^TWX||_0)$ extra divisions. This suggests the sweep operation is a much faster algorithm than the classical Gauss–Jordan elimination algorithm in online computing where the new data feeds into the process sequentially. This is the major reason our proposed O-LGT method uses the sweep operator in its core computing engine.

### 3.5.2. O-LGT Framework

The O-LGT model combines LSTM, GRU, and transformer as three sequential layers into a composite RNN to predict stock prices in high-frequency trading (HFT). The LSTM layer is for capturing important information from the input data to return an accurate LSTM output. The GRU layer takes the LSTM output as the input, which is then processed in a condensed way to prevent overfitting. Then, the transformer layer takes the GRU layer output as the input, which is processed in such a way that the interaction effects between different time steps are incorporated into the output at a more granular level. Finally, the predictions are made by concatenating the transformer output with its time length information and feeding the result through a linear regression layer.

We used the instrument PyTorch to implement the O-LGT model, with the parameters and hyper-parameters being those presented in the paper. The detailed implementation steps are as follows:

1. Pre-process the data, ensuring there are no None values in the input data, and then transfer the input data into tensors using `torch.as_Tensor()`;
2. Create a class inherited from `torch.nn.Module`; initialize it with the GRU, LSTM, and transformer layers; write the forward function according to Figure 5;
3. Initialize the PyTorch optimizer and scheduler according to Table 2;
   Choose the loss function `nn.MSELoss()` and finalize the training.

**Table 2.** Tuning-parameter specification of the experiments.

| Item | Tuning-Parameter |
|---|---|
| Optimization | Adam |
| Initial Learning Rate | 0.001 |
| Exponential Linear Decay | 0.95 |
| Epoch Number | 100 |

The notation in Figure 5 corresponds to the notation in Sections 3.3 and 3.4, and the modeling principles for each layer are shown in Sections 3.3 and 3.4. We used PyTorch to integrate each layer into the final hybrid model. The model architecture schematic of O-LGT is displayed in Figure 5 and is explained in the following.

In the initial step, the model inputs $X_T$, the matrix of observed features of the LOB data for the previous $T$ moments, into an LSTM layer, and the output of this layer, $U_T$, is stored. This output is then input into the next layer, a GRU layer, and the output of this layer, $V_T$, is stored. The output of the GRU layer is then fed into a transformer layer, and the output, $\mathbf{w}_T$, is stored. This output and time length information are concatenated into the final linear regression layer to make the final predictions.

1. Input layer: $X_T$ (matrix of observed LOB features for the previous $T$ moments);
2. LSTM layer: output $U_T$ (capture important information from the input data);
3. GRU layer: output $V_T$ (process in a condensed way to prevent overfitting);

4. Transformer layer: output $\mathbf{w}_T$ (incorporating interaction effects between different time steps);
5. Concatenation: combine transformer output with time length information;
6. Linear regression layer: make final predictions $y_T$.

In the acceleration steps, the O-LGT model uses a similar process but with the added input of the previous moment's output for each layer.

1. Input layer: $\mathbf{x}_T$ (vector of observed LOB features at time $T$) and $\mathbf{u}_{t-1}$ (previous LSTM layer output);
2. LSTM layer: output $\mathbf{u}_t$ (updated LSTM output);
3. GRU layer: output $\mathbf{v}_t$ (updated GRU output), using $\mathbf{u}_t$ and previous GRU layer output, $\mathbf{v}_{t-1}$;
4. Transformer layer: output $\mathbf{w}_t$ (updated transformer output), using $\mathbf{v}_t$ and previous transformer layer output, $\mathbf{w}_{t-1}$;
5. Concatenation: combine updated transformer output with time length information;
6. Linear regression layer: make latest prediction $y_t$.

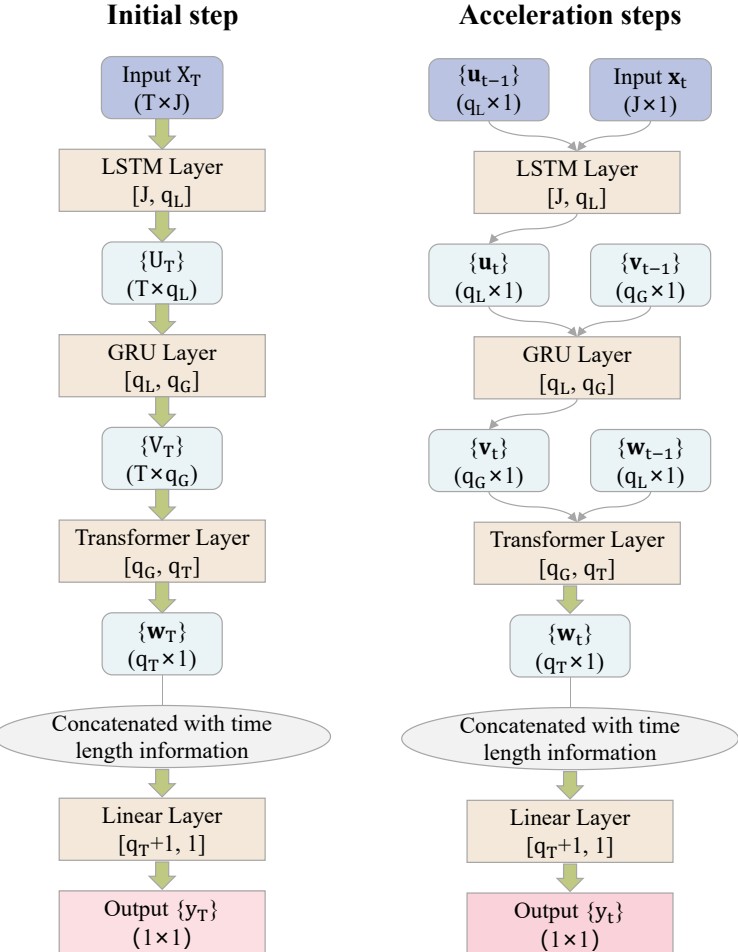

**Figure 5.** Schematic of O-LGT model architecture. The left panel shows the details of the prediction steps for $y_T$ at initial step and the right panel shows the details of the prediction steps for $y_t$ at acceleration steps. $T$ represents the length of time for the input in the initial step; $J$ represents the input dimension; $q_L, q_G, q_T$ represent the hidden layer dimensions of the LSTM, GRU, and transformer, respectively. The values of $[\cdot, \cdot]$ represent input dimensions and output dimensions. The values of $(\cdot \times \cdot)$ represent dimensions of a matrix and a vector.

Our innovative implementation setting for O-LGT was shown to have the capacity to accelerate the computation of O-LGT without compromising its accuracy. In addition, the structure of each neural network layer in O-LGT is easy to understand, train, and compute on its own.

### 3.5.3. Experimental Design for Implementation

In an HFT market, the LOB data arrives at a very high frequency (every 3 s in the case of the Chinese HFT market). It is both computationally and logistically impractical to predict stock prices for the time steps immediately after the current time step when the latest LOB data arrives. Thus, we propose to predict the prices (in terms of price percentage change) using O-LGT for time step $T + h$ when standing at the current time step $T$, where $h = 100$ (i.e., 5 min) was chosen for our Chinese market case study. On the other hand, there is no need to use the LOB data from all past time steps until the current time step $T$ to predict the prices for time step $T + h$, because the stock price dynamics manifest an LSTM behavior. This behavior is fully utilized by our O-LGT framework in that, when the current time step is between $T$ and $T + b$ (inclusive), we only use the LOB data from time step $T$ back to time step $T - s$ to make predictions for time steps $T + h$ until $T + h + b$. In this way, the required feature input data for processing O-LGT at a time step between $T$ and $T + b$ is of a time length not more than $s + b + 1$. It, thus, requires a very limited amount of computer memory, resulting in a significant acceleration of the predicting process with O-LGT. We term the implementation procedure just described for O-LGT the moving window-based prediction technique, with the back, current, and future window sizes being $s$, $b$, and $h$, respectively. For the Chinese HFT market case study, we chose $s = 99$ (4 min 57 s), $b = 19$ (57 s), and $h = 100$ (5 min). This moving window technique is illustrated in Figure 6, where the blue box represents the input features of the model and the red box represents the output of the model.

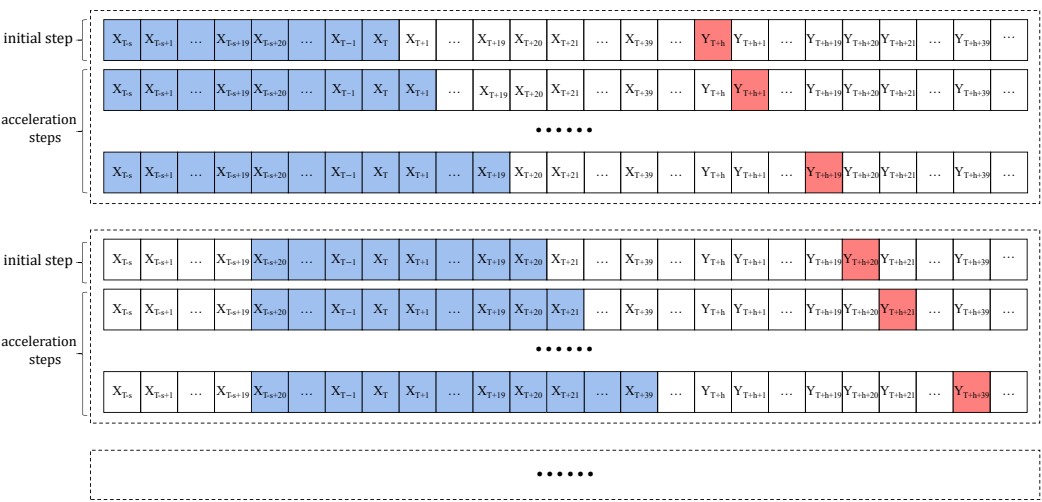

**Figure 6.** The adjusted moving window method process. Here, $s = 99$, $b = 19$, and $h = 100$. Observations of LOB data are updated every 3 s. The operation in the dashed box is updated every 20 time units (1 min). The dashed box indicates an acceleration process, where $\mathbf{x}_t$ indicates the feature information at time $t$ and $y_{T+h}$ represents the predicted stock price percentage change at time $T + h$. The first line in the first dashed box indicates that the stock price prediction $y_{T+h}$ at time $T + h$ is obtained using the feature information from time $T - s$ to time $T$. The last line in the first dashed box indicates that the stock price prediction $y_{T+h+19}$ at time $T + h + 19$ is obtained using the feature information from time $T - s$ to time $T + 19$.

Our O-LGT method also ensures a higher prediction accuracy than the other RNN methods. Denote $\mathbf{x}_{c:d}$ as the sequence of the input feature data from time $c$ to time $d$ for a stock. Given $\mathbf{x}_{1:T}$ of arbitrary time length $T$ and the future window size $h$, the online

sequential way that the target variable $y_{T+h}$ is predicted by the O-LGT model suggests $y_T$ can be formulated as follows:

$$y_{T+h} = F\left(\mathbf{x}_{(T-s):T}, \tilde{\mathbf{x}}_{T-s}\right), \tag{9}$$

where $F(\cdot, \cdot)$ is a generic function, $\tilde{\mathbf{x}}_{T-s}$ is the summary information obtained from executing O-LGT at time $T - s$, and $s$ is the back window size. Here, the back window size $s$ cannot be too small, otherwise, the result is easy to underfit, and $s$ cannot be too large, otherwise, there is not enough space for storage. Typically, we selected the appropriate $s$ according to the performance of the training data. It can be observed that the O-LGT processing of the input data sequence follows a Markov dependence pattern. Only the latest input data block $\mathbf{x}_{(T-s):T}$ at time $T$ and the previously aggregated information $\tilde{\mathbf{x}}_{T-s}$ are related to the prediction $y_{T+h}$ at the current time moment $T$. Hence, it is reasonable to conclude that an optimal value of $s$ can be found based on the training data so that the O-LGT algorithm with this optimal $s$ value will produce highly accurate predictions. However, we acknowledge that the optimal $s$ value determined this way may be too large to impact the computing complexity and speed. Fine-tuning the back window size $s$ based on trader's experience is still important and even necessary.

### 3.5.4. Data Standardization and Transformation

Recall that an RNN is characterized by the fact that all the data at different moments share the same structure and coefficient specifications. Therefore, when the model inputs are of the same length, the data will follow the same distribution at $T = 1, 2, \cdots$. However, for some variables with cumulative effects over time, such as current prices, the cumulative effects result in different variances. For example, the price at moment $t = 95$ is different from that at moment $t = 120$. Therefore, for variables with cumulative time effects, we transform the input from the absolute value to the percentage change of the current moment with respect to the previous moment. Therefore, the transformed variable will follow exactly the same distribution regardless of the time length. This treatment is more consistent with RNNs sharing the same network structure at different time lengths.

## 4. Experiment

In this paper, several experiments are presented on the CSI Smallcap 500 Index (CSI-500) in China to demonstrate the performance of the O-LGT model in stock market forecasting.

### 4.1. Data Pre-Processing for CSI-500

The dataset and the involved features are described in Section 2. Since stock prices are influenced by various factors that are not known to us, the collected data appear to contain a large amount of noise and fluctuations that are actually determined by these unknown factors. If the original stock data are directly entered into the modeling and analysis process without data pre-processing, it may reduce the accuracy of the results significantly. To increase the prediction accuracy, an appropriate data pre-processing step is necessary, as shown below.

$$\widetilde{P}(t) = \frac{p(t) - p(t-1)}{p(t-1)} \tag{10}$$

$$\widetilde{p_{\mathrm{ask}^{(k)}}}(t) = p_{\mathrm{ask}}^{(k)}(t) - p(t), \quad k = 1, 2, \ldots, 5 \tag{11}$$

$$\widetilde{p_{\mathrm{bid}^{(k)}}}(t) = p_{\mathrm{bid}}^{(k)}(t) - p(t), \quad k = 1, 2, \ldots 5 \tag{12}$$

$$\widetilde{v_{\mathrm{ask}}}(t) = \frac{v_{\mathrm{ask}}(t)}{v_0} \tag{13}$$

$$\widetilde{v_{\text{bid}}}(t) = \frac{v_{\text{bid}}(t)}{v_0} \qquad (14)$$

To prevent excessive data volatility, Equations (10)–(14) were also processed, with $v_0$ denoting the day's opening volume on the stock market. This keeps the data regularized and maintains the same distribution for the same variable.

### 4.2. Setting and Specification

In the experiments, computations by the O-LGT model were performed under the Python environment and the PyTorch framework. The Adam optimization method was used with a learning rate of 0.001 and an exponential linear decay of 0.95 after each epoch. The model's parameters were updated based on the gradients of the mean square error loss function, and the model was trained for 100 epochs using the training dataset. Table 2 lists the values of the tuning-parameters used in the model, while Table 3 lists the software and hardware configurations used for the experiments.

**Table 3.** Hardware and software configurations used in the experiments.

| Item | Configuration |
| --- | --- |
| Python Version | Python 3.9 |
| Pytoch Version | 1.10.2 |
| CPU | i7-7500U 2.70 GHz |
| RAM | 16 G |

### 4.3. Prediction Error Evaluation

The best prediction for stock prices was achieved by minimizing the loss function, which calculates the total differences between the predicted and the true price values. The resultant minimum loss provides a measure of prediction accuracy. Commonly used loss functions include mean squared error (MSE) and mean absolute error (MAE), which are defined below.

- Mean Squared Error (MSE):

$$L_{mse} = \frac{1}{N} \sum_{i=1}^{N} \sum_{t=1}^{n} (y_{it} - \hat{y}_{it})^2 \qquad (15)$$

- Mean Absolute Error (MAE):

$$L_{mae} = \frac{1}{N} \sum_{i=1}^{N} \sum_{t=1}^{n} |y_{it} - \hat{y}_{it}| \qquad (16)$$

where $\hat{y}_{it}$ denotes the predicted return of rate for stock $i$ at time $t$; $y_{it}$ denotes the true return of rate for stock $i$ at time $t$; $N$ is the number of stocks; $n$ is the number of time steps. It is easy to see that a small MSE or MAE value corresponds to a high accuracy prediction.

### 4.4. Implementation Details

In this section, we present details of applying the O-LGT approach for analyzing the Chinese LOB data. The dataset includes 100 stocks from the CSI Smallcap 500 Index market for a total of one month. As per Section 3.1, we set $y_T = \frac{p(T) - p(T-h)}{p(T-h)}$, where $h = 100$ and $p(T)$ represents the price of a stock at time $T$. In addition, recall that $\{\mathbf{x}_t, t = 1, \cdots, T\}$ is a $J$-dimensional vector time series of the LOB features with $\mathbf{x}_t = \{x_{1t}, x_{2t}, \cdots, x_{Jt}\}^\top$ and $J = 26$.

The left panel of Figure 7 gives a flow chart of the modeling and prediction process for $y_T$ at time $T$ for the initial step. The right panel is for the acceleration steps, where the back, current, and future window size of the associated moving window method are $s = 99$, $b = 19$, and $h = 100$, respectively.

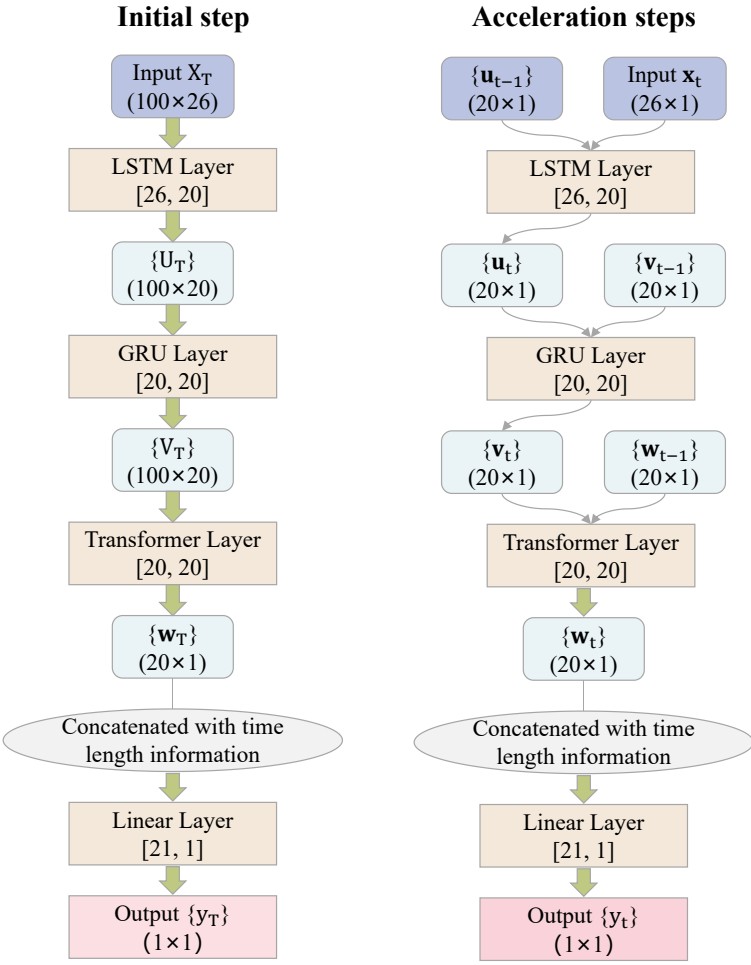

**Figure 7.** Schematic of the O-LGT structure for analyzing the Chinese LOB data. The **left** panel shows the details of the prediction steps for $y_T$ in the initial step and the **right** panel shows the details of the prediction steps for $y_t$ in the acceleration steps.

This demonstrates that, in the initial step, when time is at $T$, $X_T = [\mathbf{x}_{T-99}, \cdots, \mathbf{x}_T]^\top$—a $100 \times 26$ matrix of feature observations in all previous 100 time moments—is the input data we need for O-LGT. The input and output of the LSTM layer are both 20 dimensional. The GRU layer and the multi-head attention transformer layer are also 20 dimensional for both their input and output. The output of the transformer layer $\mathbf{w}_T$ and the information with the length of time are concatenated together, i.e., 21 dimensional. This is then loaded as the input into the linear layer. The final output is the prediction of the stock price percentage change value $y_{T+100}$.

In the acceleration steps when $T < t < T + 20$, $\mathbf{x}_t$ has $J = 26$ features in time $t$; $\mathbf{u}_{t-1}$, $\mathbf{v}_{t-1}$, and $\mathbf{w}_{t-1}$ are $20 \times 1$ vectors at time $t - 1$; $\mathbf{u}_t$, $\mathbf{v}_t$, and $\mathbf{w}_t$ are $20 \times 1$ vectors at time $t$. The dimensions of the inputs and outputs are the same for each layer and the final output is the prediction of the stock price percentage change value $y_{t+100}$.

Regarding the input of O-LGT, recall that we have 4740 time steps for each stock every trading day and there were 22 trading days in November 2021 in the Chinese HFT market. LOB information from every segment of 100 (i.e., $s = 99$) plus up to 19 (i.e., $b = 19$) consecutive time steps is used as a single input, and the rate of return after 5 min (i.e.,

$h = 100$) is the corresponding output. For prediction in each time step $t \in [T, T + b]$, the feature data $\mathbf{x}_{(T-s):t}$ is used as the input, where $T$ is a sharp minute time step (e.g., 9 h:50 m:00 s:am sharp), $b = 19$ (57 s), and $s = 99$ (4 min 57 s). Thus, we have around 4610 consecutive and overlapping segments of input data of the form $\mathbf{x}_{(T-s):t}$ on every trading day. Instead of testing the performance of O-LGT on all these 4610 segments of input data on each trading day (which is referred to as fixed testing), we were able to do it on a stratified random sample from these 4610 segments. For example, one such random sample could be obtained by first partitioning the 4610 segments into 461 consecutive sections and then randomly choosing a segment from each section. This testing, based on random sampling, is referred to as random testing. A possible advantage of using random testing is the reduced auto-correlations in the random sample. Instead of using a fixed back window size $s$ for training the model, one could use a randomly selected $s$ value at each updating step for the training. For example, $s$ could be randomly selected from the interval $[89, 109]$. Selecting $s$ at random is referred to as random training.

*4.5. Experiment Results*

In this section, we report the results of experiments to demonstrate the performance of our O-LGT model. We performed experimental studies for the CSI-500 dataset under three setups: Setting I, Setting II, and Setting III. Setting I simulated three comparative experiments specified by the fixed/random training/testing combinations. Setting II simulated a missing value scenario, under which O-LGT was compared with LGT and the linear model. Setting III was a scenario with no missing values in the input data, under which O-LGT was compared with XGBoost, DeepLOB, and DeepAcc, in addition to the linear model and LGT.

First, to confirm the validity of the experimental design, we carried out three comparison experiments in Setting I:

- The moving window for model training had a fixed back section size $s = 99$. The model testing was performed on all 4610 input segments;
- The moving window for model training had a fixed back section size $s = 99$, while the model testing was performed on a stratified random sample of 461 input segments;
- The moving window for model training had its back section size $s$, which was chosen from $[89, 109]$ at random, while the model testing was performed on a stratified random sample of 461 input segments.

Table 4 demonstrates that the performances of the O-LGT method were similar to each other under the three experiments, verifying the validity of our design. There was no significant difference in MSE and MAE when the input lengths of both the training and test sets were randomized compared to when the input lengths of both the training and test sets were fixed.

**Table 4.** Results of the three comparison experiments to verify the validity of the model design. Values in () represent standard errors. ↑ means the higher the value, the better the model performs. ↓ means the lower the value, the better the model performs.

| Methods | $100 \times$ **RMSE** ↓ | $100 \times$ **MAE** ↓ |
|---|---|---|
| Training fixed -> Test fixed | 0.171 (0.0466) | 0.0943 (0.00454) |
| Training fixed -> Test random | 0.174 (0.0487) | 0.0955 (0.00471) |
| Training random -> Test random | 0.171 (0.0486) | 0.0948 (0.00468) |

Next, we considered the scenario with missing values in Setting II. Specifically, values at five time steps in each segment of input data were removed at random. Table 5 shows that when there were missing values, the prediction performance of LGT and O-LGT was much better than that of the linear model. Comparing LGT and O-LGT, we found no significant difference between MSE and MAE. The superiority of O-LGT in the presence of missing values was determined by its design features, as it was more flexible in the choice

of input length, meaning the prediction accuracy was unaffected even in the presence of missing values.

    In the absence of missing values in Setting III, we compared the O-LGT with not only the traditional models mentioned above, i.e., the linear model and GRU, but also with the DeepLOB and DeepAcc models reviewed in Section 1, which are two powerful hybrid models with a superior overall performance. Table 6 demonstrates that O-LGT and LGT had clear advantages over the other comparative models in terms of prediction accuracy. In terms of the computing time, O-LGT was about 38 times faster than LGT, about 64 times faster than DeepLOB, and about 12 times faster than DeepAcc. This shows that our O-LGT model can significantly speed up prediction while maintaining prediction accuracy, which is very beneficial for early risk assessment in the stock market.

**Table 5.** Benchmark models comparison in missing value scenarios. Values in () represent standard errors. ↑ means the higher the value, the better the model performs. ↓ means the lower the value, the better the model performs.

| Methods | $100 \times$ RMSE ↓ | $100 \times$ MAE ↓ |
| --- | --- | --- |
| Linear | 0.352 (0.1300) | 0.2290 (0.12900) |
| LGT | 0.171 (0.0467) | 0.0944 (0.00454) |
| O-LGT | 0.171 (0.0486) | 0.0948 (0.00468) |

**Table 6.** Performance comparison of different models with no missing values. Values in () represent standard errors. ↑ means the higher the value, the better the model performs. ↓ means the lower the value, the better the model performs.

| Methods | $100 \times$ RMSE ↓ | $100 \times$ MAE ↓ | Time (Millisecond) |
| --- | --- | --- | --- |
| Linear | 0.330 (0.1160) | 0.2030 (0.01180) | 0.0352 |
| XGBoost | 0.259 (0.0880) | 0.1530 (0.00754) | 0.704 |
| DeepLOB | 0.173 (0.0468) | 0.0945 (0.00457) | 3.68 |
| DeepAcc | 0.178 (0.0466) | 0.0930 (0.00454) | 0.695 |
| LGT | 0.171 (0.0465) | 0.0943 (0.00454) | 2.21 |
| O-LGT | 0.171 (0.0486) | 0.0948 (0.00468) | 0.0579 |

    In summary, our O-LGT model has the capacity to quickly and accurately predict stock price in high-frequency trading markets. The results confirm the validity of the experimental design and demonstrate the superior performance of O-LGT in terms of both prediction accuracy and computational speed compared to other models.

## 5. Conclusions

    In this study, we developed O-LGT, an online hybrid recurrent neural network model tailored for analyzing LOB data and predicting stock price fluctuations in a high-frequency trading (HFT) environment. The O-LGT model combines LSTM, GRU, and transformer layers, and features efficient storage management, enabling rapid computation while maintaining high prediction accuracy and feasible memory usage. Our experimental results on the CSI-500 dataset confirmed the validity of our experimental design and demonstrated the superior performance of O-LGT in terms of both prediction accuracy and computational speed in comparison with other network integration models, such as LGT, DeepLOB, and DeepAcc. We addressed the often-overlooked aspect of computation speed in high-frequency trading, providing traders with a significant advantage in HFT environments by enabling faster decision-making and order execution. Specifically, it shows that, in most cases, our model achieves a similar speed but with a much higher accuracy than the conventional fast supervised learning models for HFT. On the other hand, with a slight sacrifice in accuracy, O-LGT is approximately 12 to 64 times faster than the existing high-accuracy neural network models for the LOB data in the Chinese market.

Future work can focus on further improving O-LGT's performance and generalizability, and exploring its applications in other financial markets with high-frequency data and its performance in predicting other financial instruments.

**Author Contributions:** Conceptualization, C.L. and G.Q.; methodology, C.L., L.S. and G.Q.; software, C.L. and L.S.; validation, G.Q. and C.L.; formal analysis, C.L. and L.S.; investigation, C.L., L.S. and G.Q.; data curation, C.L.; writing—original draft preparation, C.L. and L.S.; writing—review and editing, C.L., L.S. and G.Q.; visualization, L.S.; supervision, G.Q. All authors have read and agreed to the published version of the manuscript.

**Funding:** This research received no external funding.

**Acknowledgments:** We thank the two anonymous reviewers for valuable comments which helped us to improve quality of the manuscript and its presentation.

**Conflicts of Interest:** The authors declare no conflict of interest.

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
