# Peer review of "Online Hybrid Neural Network for Stock Price Prediction: A Case Study of High-Frequency Stock Trading in the Chinese Market"

_econometrics, doi:10.3390/econometrics11020013_

Round 1

Reviewer 1 Report

I think the paper is quite felicitous and wish the authors good luck in this process. I have elaborated on some questions in the attached PDF. However, as indicated in the online template, I think that the authors did a great job and I have not much to add.

Author Response

Thank you for your very positive review. We are not able to find any specific comment in your attached PDF. Here we attach our response to the other review for your reference.

Reviewer 2 Report

High-frequency trading (HFT) is a popular trading method that relies on fast and accurate predictions of stock price movements. Traditional models often struggle to keep up with the speed of HFT and fail to accurately predict the movement of stock prices. This paper presents a new online hybrid neural network model, O-LGT, that aims to overcome these limitations by using three consecutive layers of recurrent neural networks (LSTM, GRU, and Transformer) to extract information from the features of the limit order book (LOB) data. The authors compare the performance of O-LGT with several classical models and other deep learning models using the CSI-500 dataset.

Positive aspects of the paper:

  • The paper proposes a novel online hybrid recurrent neural network model (O-LGT) for predicting stock price fluctuations in high-frequency trading (HFT) environments.
  • The paper demonstrates the effectiveness of O-LGT in comparison to several other models in three different experimental setups.
  • The paper uses three consecutive layers of recurrent neural networks: LSTM, GRU, and Transformer, to extract information from the features at different stages.
  • The paper uses a missing value scenario to show that O-LGT is more accurate than the linear model, and a scenario with no missing values to compare O-LGT with XGBoost, DeepLOB, and DeepAcc.
  • The paper shows that O-LGT is computationally substantially faster than network integration models and can still achieve high accuracy rates.

However, the paper has some flaws, which should be eliminated:

  1. Systemizing variables in a table or other view can make it easier for readers to understand and follow the research.
  2. There is a lot of information about classical RNN, GRU, LSTM, but there should be more information about O-LGT. Providing more information about O-LGT, the proposed model, can help readers understand the novelty and contribution of the paper.
  3. The authors need to detail the PyTorch instruments (which were used), hyperparameters, and so on. The results of the implementation in PyTorch should be included in the appendix, which can help other researchers replicate the study and improve the credibility of the results.
  4. In paragraph 4.3, the R-squared metric does not correspond to the essence of the given metric. RMSE, MAPE, SMAPE should be used instead.
  5. The input data is highly limited to one market. Expanding the output for comparison and using different datasets can improve the generalizability of the model and strengthen the validity of the findings.
  6. Restructuring the conclusions to emphasize the results and contribution of the study can make them more compelling and informative for readers. A discussion and further development section can provide readers with insights on the implications and future directions of the research.

It is also noted that flaws 4, 5, and 6 require major revision.

Author Response

Thank you for taking your time to review our submission. Please see the attachment of our response.

Round 2

Reviewer 2 Report

The material has been revised according to the feedback provided. I recommend it for publication